# Mitochondrial stress-activated cGAS-STING pathway inhibits thermogenic program and contributes to overnutrition-induced obesity in mice

Juli Bai[1,2 ✉], Christopher Cervantes[1], Sijia He[1], Jieyu He[1,2], George R. Plasko[1], Jie Wen[1,2], Zhi Li[1], Dongqing Yin[1], Chuntao Zhang[1], Meilian Liu [2,4], Lily Q. Dong [3] & Feng Liu [1,2 ✉]

Obesity is a global epidemic that is caused by excessive energy intake or inefficient energy expenditure. Brown or beige fat dissipates energy as heat through non-shivering thermogenesis by their high density of mitochondria. However, how the mitochondrial stress-induced signal is coupled to the cellular thermogenic program remains elusive. Here, we show that mitochondrial DNA escape-induced activation of the cGAS-STING pathway negatively regulates thermogenesis in fat-specific DsbA-L knockout mice, a model of adipose tissue mitochondrial stress. Conversely, fat-specific overexpression of DsbA-L or knockout of STING protects mice against high-fat diet-induced obesity. Mechanistically, activation of the cGAS-STING pathway in adipocytes activated phosphodiesterase PDE3B/PDE4, leading to decreased cAMP levels and PKA signaling, thus reduced thermogenesis. Our study demonstrates that mitochondrial stress-activated cGAS-STING pathway functions as a sentinel signal that suppresses thermogenesis in adipose tissue. Targeting adipose cGAS-STING pathway may thus be a potential therapeutic strategy to counteract overnutrition-induced obesity and its associated metabolic diseases.

---

[1] Departments of Pharmacology, University of Texas Health San Antonio, San Antonio, TX, USA. [2] Department of Metabolism and Endocrinology and the Metabolic Syndrome Research Center, The Second Xiangya Hospital, Central South University and National Clinical Research center for Metabolic Diseases, Changsha, Hunan, China. [3] Departments of Cell Systems & Anatomy, University of Texas Health San Antonio, San Antonio, TX, USA. [4] Department of Biochemistry and Molecular Biology, University of New Mexico Health Science Center, Albuquerque, NM, USA. ✉email: BaiJ@uthscsa.edu; Liuf@uthscsa.edu

In response to viral and microbial-induced cellular stress, innate immune cells initiate an immune response to protect host from infections and restore disrupted cellular homeostasis. This immune signaling cascade is initiated by double-stranded DNA-induced activation of the cGMP–AMP (cGAMP) synthase (cGAS), which synthesizes the eukaryotic secondary messenger 2′3′-cGAMP[1]. cGAMP binds to an adapter protein named Stimulator of Interferon Genes (STING), leading to the activation of downstream kinases such as TANK Binding Kinase 1 (TBK1) or IKKε, which consequently mounts an immune response[1]. In recent years, several studies showed that in addition to being activated by pathogen-derived DNAs from viral or microbial infection, the cGAS–STING pathway could also be initiated by self-DNAs such as mitochondrial DNAs (mtDNAs) aberrantly localized in the cytosol under certain stress conditions[2–5]. This finding suggests broad functions of the innate immune pathway in mitochondrial enriched tissues or cells.

Brown adipose tissue (BAT), which possess a high density of mitochondria, plays a critical role in adaptive thermogenesis. Another type of cell, so-called brown-in-white (brite) or beige adipocyte, which is primarily induced from subcutaneous white adipocyte to become brown-like thermogenically active cells with increased density of mitochondria. In both rodents and adult humans, the presence of these thermogenic adipocytes has demonstrated a promising therapeutic potential to treat obesity and diabetes[6–8]. Thermogenesis is an energy-consuming process that is heavily dependent on β3 adrenergic receptor mediated protein kinase A (PKA) activation and its downstream mitochondrial effector uncoupling protein 1 (UCP1), which short circuits the respiratory chain to dissipate chemical energy in the form of heat. However, while mitochondria are normally considered as a "receiving-end" organelle to produce energy for the cell and to burn energy as heat, much less is known about whether and how mitochondria stress-induced signals is coupled to cellular thermogenic program to regulate energy homeostasis.

Several studies suggested that mitochondrial dysfunction, such as respiratory chain dysregulation, lower mtDNA content, and fatty acid oxidation deficiency, all lead to reduced thermogenesis or impaired beige fat development[9–13]. Conversely, improving mitochondrial function promoted brown or beige fat thermogenesis and protected against high-fat diet (HFD)-induced obesity[14,15]. However, the identity of the signal transducers between mitochondria and nuclear thermogenic program remains largely unknown. We recently found that HFD-feeding or fat-specific knockout of disulfide bond A oxidoreductase-like protein (DsbA-L), a chaperon-like mitochondrial localized protein, promoted mtDNA release into the cytosol where it activated the cGAS–STING pathway in noninnate immune cells such as adipocytes[5]. The activation of the cGAS–STING pathway promoted inflammation and exacerbated obesity or DsbA-L knockout-induced insulin resistance in mice, thereby uncovering an important mechanism linking immunity and metabolism[5,16].

In the current study, we show that cytosolic mtDNA-induced activation of the cGAS–STING pathway functions as a key sentinel signaling to suppress thermogenic fuel influx and gene expression in adipose tissue of the DsbA-L[fKO] mice. Consistent with this finding, knockout of STING or suppression of cGAS–STING pathway by fat-specific overexpression of DsbA-L increased thermogenic gene expression in mouse adipose tissue. Mechanistically, we found that the activation of the cGAS–STING pathway in brown adipocytes increased phosphodiesterase (PDE) activity, leading to reduced cellular cAMP levels and decreased PKA activity. Our study uncovers a link of mitochondrial dysfunction-derived immune response to thermogenesis and energy expenditure in adipose tissue, suggesting the

cGAS–STING pathway may be an effective therapeutic target for the treatment of obesity and its associated diseases.

## Results

**Fat DsbA-L deficiency decreases energy expenditure in mice.** We previously found that in addition to increased chronic sterile inflammation and exacerbated insulin resistance, the fat-specific DsbA-L knockout mice (DsbA-L[fKO]) also displayed increased susceptibility to HFD-induced obesity compared with the loxp littermates. Given that fat-specific knockout of DsbA-L had no effect on food intake[5] and physical activity (Fig. 1a), we postulated that the increased body weight and adiposity in the DsbA-L[fKO] mice might be due to decreased thermogenesis and energy expenditure. To test this, we first compared the expression of endogenous DsbA-L, a mitochondrial enriched protein, in mouse fat tissues. The mRNA (Fig. 1b) and protein (Fig. 1c) levels of DsbA-L are notably higher in mouse BAT than in white adipose tissue (WAT). In addition, DsbA-L expression is greatly induced during brown adipocyte differentiation (Supplementary Fig. 1a). The expression of DsbA-L was also greatly induced by treating brown adipocytes with CL316243, a β-3 adrenoceptor agonist that mimics cold stress, or by rosiglitazone, a PPARγ agonist that promotes UCP1 expression (Supplementary Fig. 1b). Consistent with our hypothesis that DsbA-L deficiency inhibits energy expenditure, both the oxygen consumption (VO₂) (Fig. 1d) and the resting metabolic rate (Supplementary Fig. 1c) were significantly reduced in fat-specific DsbA-L knockout mice (DsbA-L[fKO]) compared with floxed wild-type control mice. In addition, the DsbA-L[fKO] mice displayed a lower basal core body temperature at either regular housing conditions (Fig. 1e) or when exposed to cold (Fig. 1f), which was not due to reduced physical activity (Supplementary Fig. 1d). Fat-specific knockout of DsbA-L had no significant effect on the respiratory quotient of the mice (Supplementary Fig. 1e), indicating that DsbA-L deficiency in adipose tissue did not affect the preference of the mice to use carbohydrates or lipids as the main energy source.

**DsbA-L regulates thermogenesis in brown and beige fat.** To determine the mechanism by which DsbA-L regulates energy expenditure, we examined thermogenic gene expression in BAT and inguinal WAT (iWAT) of DsbA-L[fKO] mice and loxp control littermates. Cold exposure significantly increased the mRNA levels of DsbA-L, UCP1, C/EBPβ, and PGC1α in both BAT (Fig. 2a) and iWAT (Fig. 2b) of loxp control mice. The mRNA levels of TBX-1 and TMEM26, two well-established beige fat markers, were also enhanced in iWAT of cold exposed wild-type control mice (Fig. 2b). However, the stimulatory effect of cold exposure on these thermogenic gene expressions was significantly reduced in DsbA-L[fKO] mice (Fig. 2a, b). Consistent with reduced mRNA expression, the protein levels of UCP1, C/EBPβ, and PGC1α were also significantly reduced in BAT and iWAT of DsbA-L[fKO] mice compared with the wild-type control mice exposed to cold (Fig. 2c, d). Hematoxylin and eosin (H&E) staining experiments revealed that adipose tissue-specific knockout of DsbA-L greatly suppressed cold-induced formation of small and multilocular lipid droplets beige adipocytes in mouse iWAT and increased lipid accumulation in BAT (Fig. 2e). Suppressing DsbA-L expression in cultured brown adipocytes had no effect on cell differentiation but greatly inhibited *Ucp1, C/ebpβ, Pgc1α,* and *Prdm16* thermogenic gene expression in brown adipocytes (Fig. 2f), suggesting that DsbA-L has a cell-autonomous effect on thermogenic gene expression. Conversely, fat-specific overexpression of DsbA-L in mice markedly increased the expression levels of *Ucp1, Pgc1α,* and *C/ebpβ* in both BAT (Fig. 2g) and iWAT (Fig. 2h), which is consistent with our

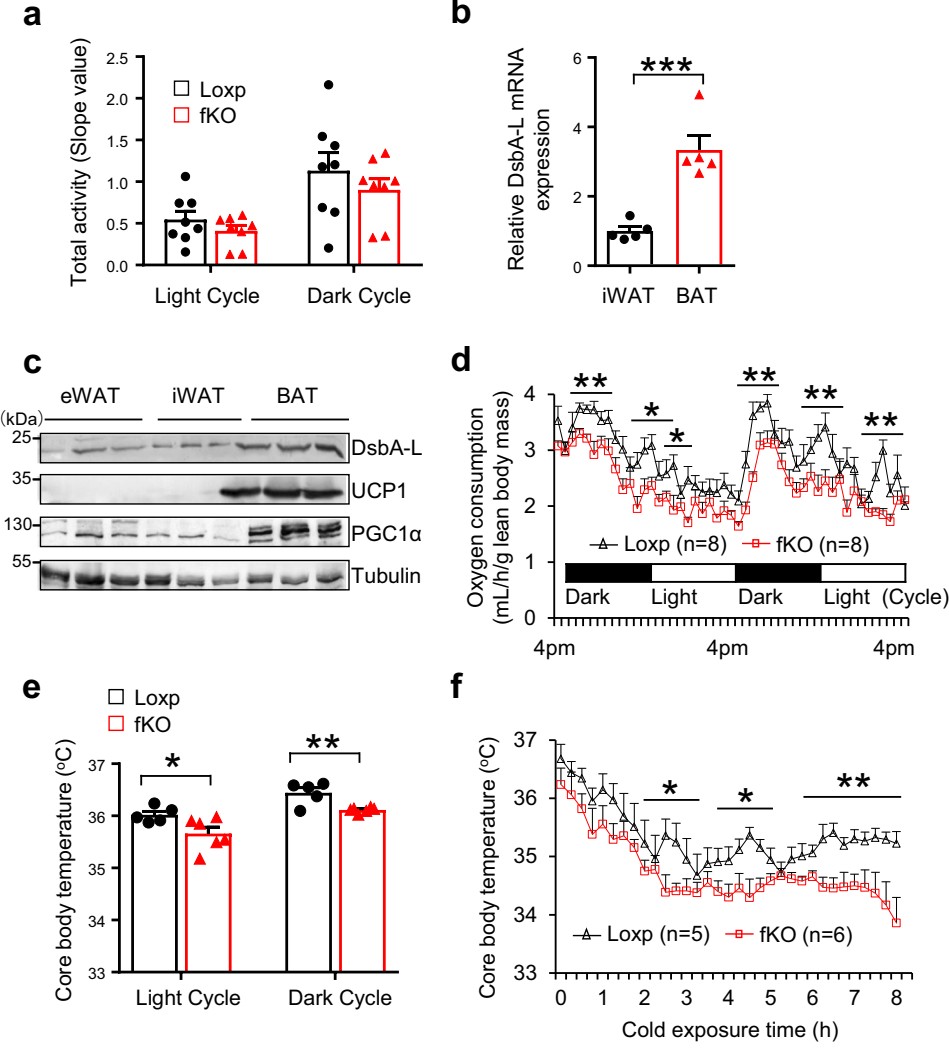

**Fig. 1 Fat-specific knockout of DsbA-L reduced energy expenditure in mice. a** The activities of DsbA-L[fKO] ($n = 8$) and loxp control ($n = 8$) mice was measured during a 48-h period, including two light/dark cycles. **b** The mRNA levels of DsbA-L in inguinal WAT (iWAT) ($n = 5$) and BAT ($n = 5$) were determined by qPCR and normalized to β-actin. **c** Immunoblot analysis of DsbA-L, UCP1, PGC1α expression in epididymal WAT (eWAT), iWAT and BAT from C57BL/6 wild-type mice ($n = 3$). **d** Oxygen consumption of DsbA-L[fKO] and loxp control mice was measured during a 48-h period, including two light/dark cycles. **e** Core body temperature of DsbA-L[fKO] ($n = 5$) and loxp control ($n = 6$) mice at room temperature (24 °C) in the feeding condition. **f** Core body temperature of DsbA-L[fKO] and loxp control mice exposed to cold (4 °C) in the feeding condition at different time points as indicated. Data are presented as mean ± SEM of biologically independent samples, $^*p < 0.05$, $^{**}p < 0.01$, and $^{***}p < 0.001$ by unpaired two-tailed t-test.

previous finding that fat-specific overexpression of DsbA-L enhanced energy expenditure and protected mice from HFD-induced obesity[17]. Several studies reveal the presence of UCP1-independent mechanisms to promote beige fat thermogenesis, including creatine-driven substrate cycle[18] and sarco/endoplasmic reticulum (ER) $Ca^{2+}$-ATPase 2b (SERCA2b)-mediated calcium cycle[19]. However, we found that cold exposure had a comparable stimulatory effect on the mRNA expression of calcium cycle-related gene *serca2b* and creatine metabolism-related genes including *ckmt1, ckmt2, gatm* in iWAT of both the loxp control mice and DsbA-L[fKO] mice (Supplementary Fig. 2a, b, c, d), indicating DsbA-L deficiency in adipose tissue had no significant effect on these UCP1-independent mechanisms underlying cold-induced beige fat thermogenesis.

**DsbA-L promotes lipolysis and fatty acid oxidation**. To determine the mechanism by which fat-specific knockout of DsbA-L increased adiposity in mice, we examined the expression of genes involved in lipolysis and fatty acid oxidation in BAT and iWAT.

The mRNA, phosphorylation, and protein levels of key lipolytic enzymes such as hormone-sensitive lipase (HSL) and adipose triglyceride lipase (ATGL) were significantly reduced in both BAT and iWAT of DsbA-L[fKO] mice compared with wild-type littermates (Fig. 3a, b and Supplementary Fig. 3a, b). Concurrently, the β3 adrenoceptor agonist isoproterenol-stimulated glycerol and fatty acid release were both significantly suppressed in primary brown and subcutaneous adipocytes isolated from DsbA-L[fKO] mice compared with their control mice (Fig. 3c–f). A significant reduction in CL316243-induced lipolysis was also observed in DsbA-L-suppressed brown adipocytes (Supplementary Fig. 3c, d), concurrently with increased lipid droplet formation (Supplementary Fig. 3e). In addition to decreased lipolysis, a significant reduction in the expression of genes involved in fatty acid oxidation, such as citrate transport protein 1, medium-chain acyl-CoA dehydrogenase, peroxisome proliferator-activated receptor α, and 3-hydroxy-3-methylglutaryl-CoA synthase 2, was also observed in BAT and iWAT of DsbA-L[fKO] mice compared with floxed control mice (Fig. 3g, h).

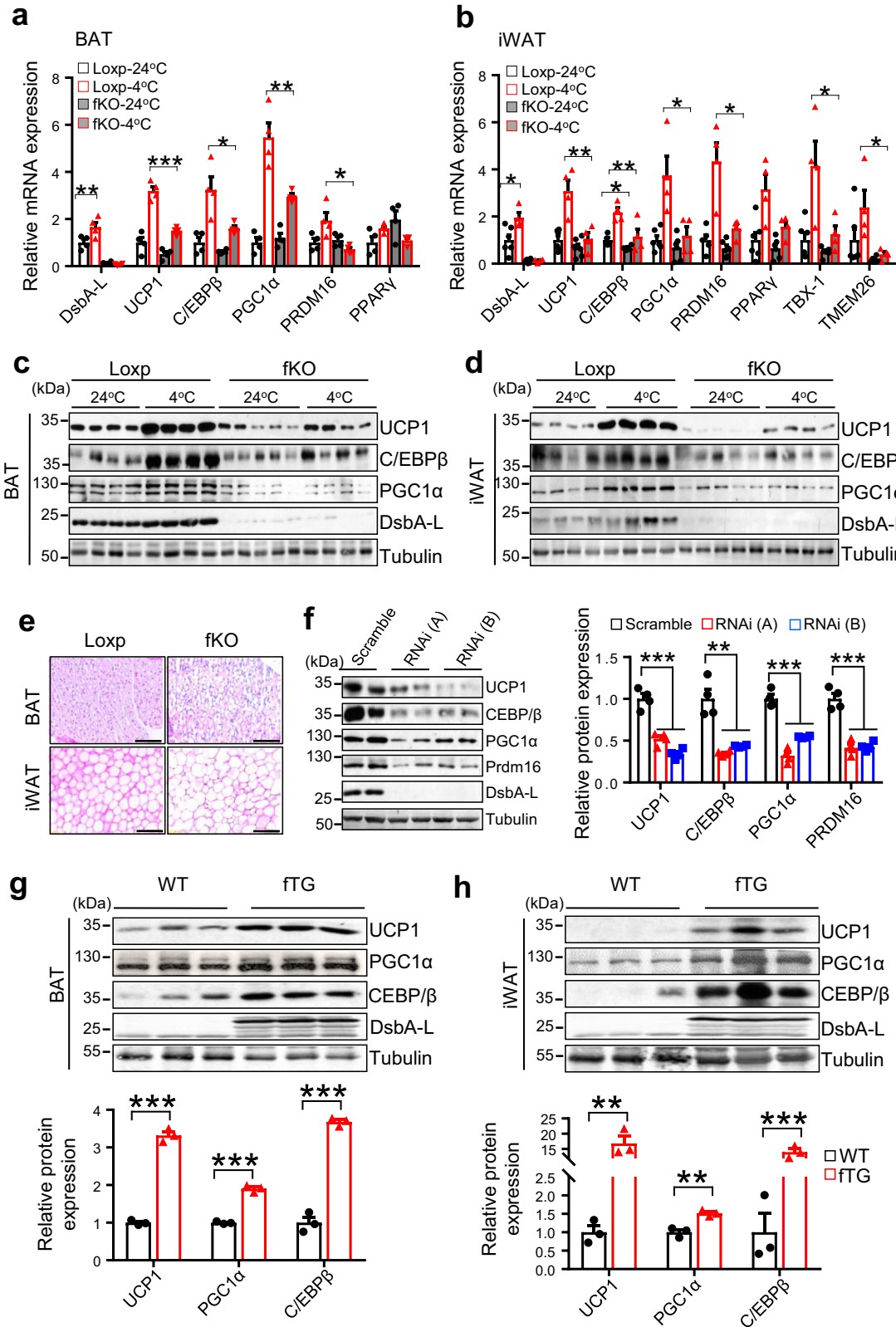

Consistently, palmitic acid-induced fatty acid oxidation was significantly decreased in primary adipocytes isolated from brown and subcutaneous white fat of DsbA-L$^{fKO}$ mice compared with the control mice (Fig. 3i, j). In contrast, both mRNA and protein levels of lipogenic genes, such as sterol regulatory element-binding protein, fatty acid synthase (FAS), and acetyl-CoA carboxylase (ACC), were significantly increased in iWAT and BAT

in DsbA-L$^{fKO}$ mice compared with their control mice (Supplementary 3a, b, f, g). However, no significant difference in the expression levels of genes involved in lipid uptake such as CD36, lipoprotein lipase fatty acid transport protein, and adipogenesis marker FABP4 was observed between DsbA-L$^{fKO}$ and the floxed control mice (Supplementary Fig. 3h–j). On the other hand, primary adipocytes from BAT and iWAT of DsbA-L$^{fTG}$ mice

**Fig. 2 DsbA-L is positively correlated with thermogenic gene expression in brown and beige fat.** Cold exposure-induced mRNA expression in **a** BAT and **b** iWAT of DsbA-L$^{fKO}$ (24 °C, $n = 6$; 4 °C, $n = 4$) and loxp control (24 °C, $n = 4$; 4 °C, $n = 4$) mice. The mRNA levels were determined by qPCR and normalized to β-actin. Immunoblot analysis for cold exposure-induced protein expression of UCP1, C/EBPβ, PGC1α, and DsbA-L in **c** BAT and **d** iWAT of DsbA-L$^{fKO}$ and loxp control mice ($n = 4$ for each group). **e** Representative H&E stain of BAT and iWAT of DsbA-L$^{fKO}$ and loxp control mice. Scale bar: 200 μM. **f** Immunoblot analysis for protein expression of thermogenic genes in DsbA-L RNAi-suppressed stable brown adipocytes and control scramble cells ($n = 4$ for each group). The data were semiquantified by ImageJ program. Immunoblot analysis for protein expression of thermogenic genes in **g** BAT and **h** iWAT of fat-specific DsbA-L transgenic mice (DsbA-L$^{fTG}$) ($n = 3$) and their control littermates ($n = 3$). The data were semiquantified by ImageJ program. Data are presented as mean ± SEM of biologically independent samples, *$p < 0.05$, **$p < 0.01$, and ***$p < 0.001$ by unpaired two-tailed $t$-test (for comparison between two groups) or one-way ANOVA (for comparison of multiple groups).

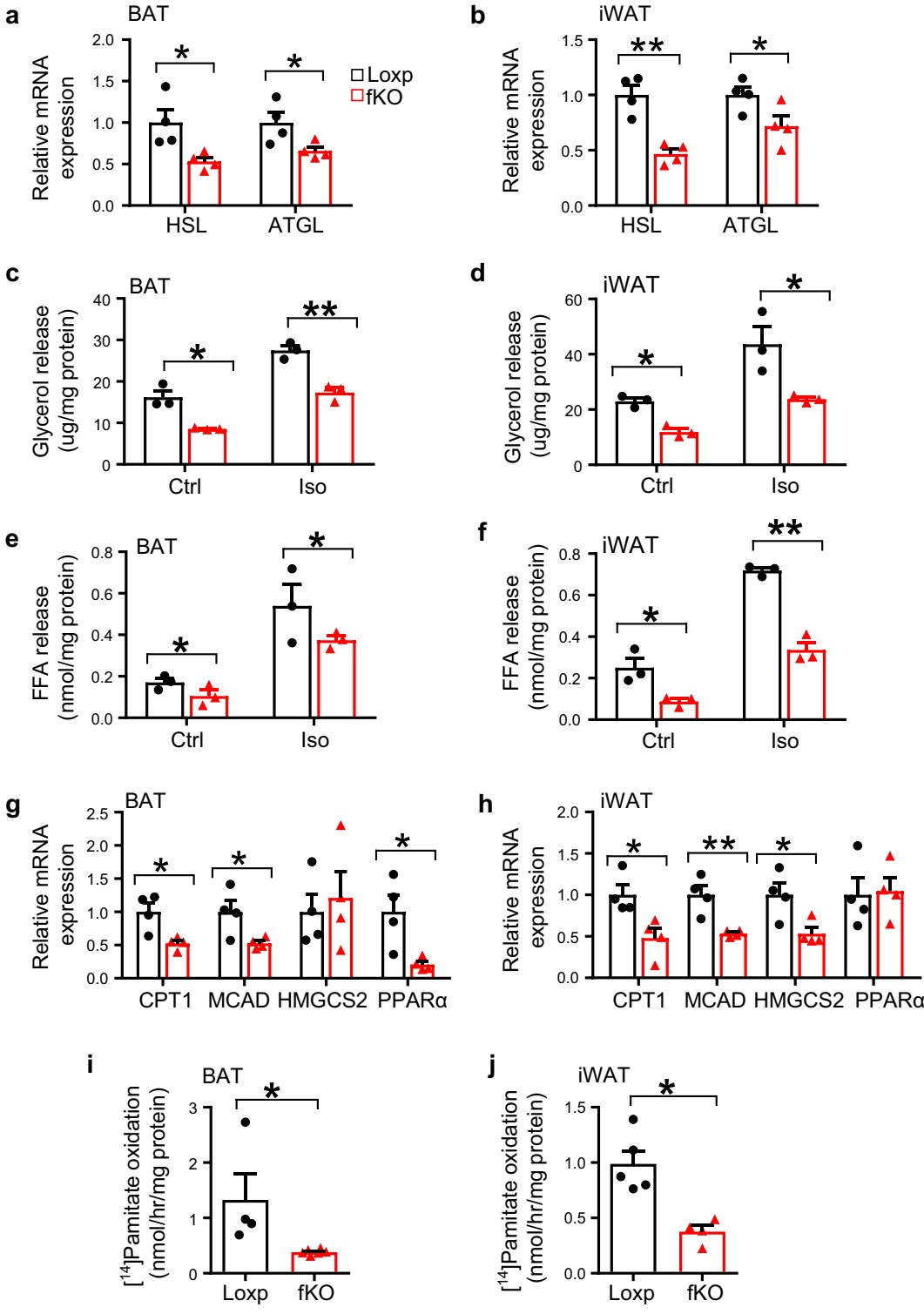

**Fig. 3** (continued)

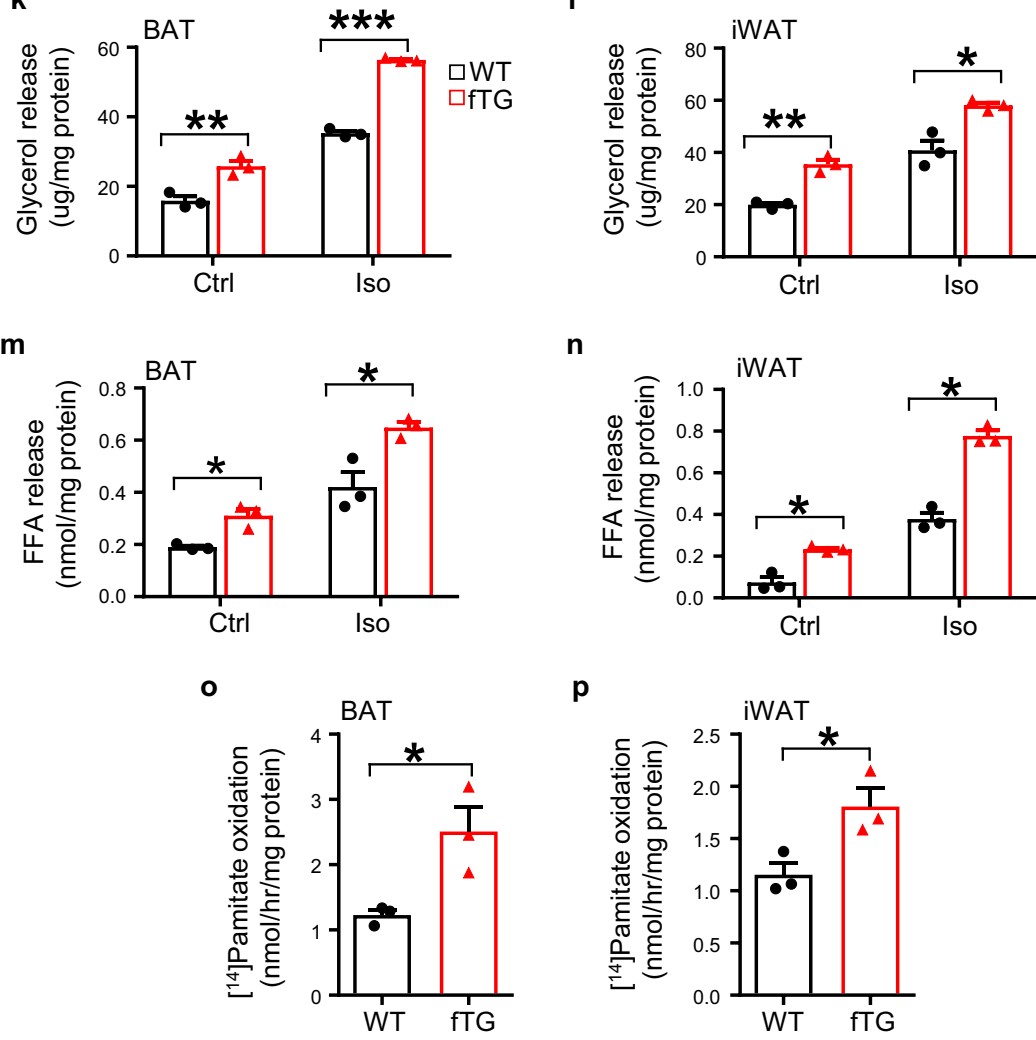

**Fig. 3 DsbA-L promotes lipolysis and fatty acid oxidation in adipose tissue.** The mRNA levels of lipolytic genes in **a** BAT and **b** iWAT of DsbA-L[fKO] ($n = 4$) and loxp control ($n = 4$) mice were determined by qPCR and normalized to β-actin. Glycerol release from primary-cultured adipocytes of **c** BAT and **d** iWAT of DsbA-L[fKO] ($n = 3$) and loxp control ($n = 3$) mice. Free fatty acid release from primary-cultured adipocytes of **e** BAT and **f** iWAT of DsbA-L[fKO] ($n = 3$) and loxp control ($n = 3$) mice. The mRNA levels of fatty acid oxidation-related genes in **g** BAT and **h** iWAT of DsbA-L[fKO] ($n = 4$) and loxp control ($n = 4$) mice were determined by qPCR and normalized to β-actin. The oxidation of [14]C-palmitate in **i** BAT and **j** iWAT from DsbA-L[fKO] ($n = 4$ for BAT and $n = 5$ for iWAT) and loxp ($n = 6$ for BAT and $n = 4$ for iWAT) control mice. Glycerol release from primary-cultured adipocytes of **k** BAT and **l** iWAT of DsbA-L[fTG] and their control mice ($n = 3$ for each group). Free fatty acid release from isolated **m** BAT and **n** iWAT of DsbA-L[fTG] and their control mice ($n = 3$ for each group). The oxidation of [14]C-palmitate in **o** BAT and **p** iWAT isolated from DsbA-L[fTG] and their control mice ($n = 3$ for each group). Data are presented as mean ± SEM of biologically independent samples, *$p < 0.05$, **$p < 0.01$, and ***$p < 0.001$ by unpaired two-tailed $t$-test.

displayed enhanced β3 adrenoceptor agonist-stimulated glycerol and fatty acid release compared with their wild-type control mice (Fig. 3k–n). Furthermore, fatty acid oxidation was also significantly increased in primary adipocytes isolated from BAT (Fig. 3o) and iWAT (Fig. 3p) of DsbA-L[fTG] mice compared with wild-type littermates. Taken together, these findings reveal that DsbA-L is a key regulator of lipolysis and fatty acid oxidation.

**The cGAS–STING pathway inhibits thermogenic gene expression.** We previously found that fat-specific knockout of DsbA-L activated the cGAS–STING signaling pathway in WAT[5,16]. Given that DsbA-L is highly enriched in BAT, we asked whether the cGAS–STING pathway is activated in BAT of the DsbA-L[fKO] mice and whether the activation plays a causal role in reduced thermogenesis in the mice. Indeed, cytosolic mtDNA

release was dramatically increased in BAT of the DsbA-L[fKO] mice compared with the wild-type control mice (Fig. 4a). Increased mtDNA release was also observed in primary-cultured brown adipocytes from DsbA-L[fKO] mice compared with those from wild-type control mice (Supplementary Fig. 4a), demonstrating a cell-autonomous role of DsbA-L in regulating mtDNA release. Consistent with these findings, the phosphorylation of TBK1 as well as the expression of cGAS, STING, and TNFα were greatly increased in BAT of the DsbA-L[fKO] mice compared with the wild-type control mice, which is correlated with reduced UCP1 expression (Fig. 4b). Suppressing DsbA-L expression also reduced the expression of UCP1 and increased the phosphorylation of TBK1 and IRF3 in brown adipocytes (Fig. 4c), concurrently with increased 2′3′-cGAMP levels (Fig. 4d) and mtDNA release into the cytosol (Supplementary Fig. 4b–d). In contrast, stably over-expression of DsbA-L restored UCP1 expression in DsbA-L-deficient brown adipocytes, which is correlated with reduced

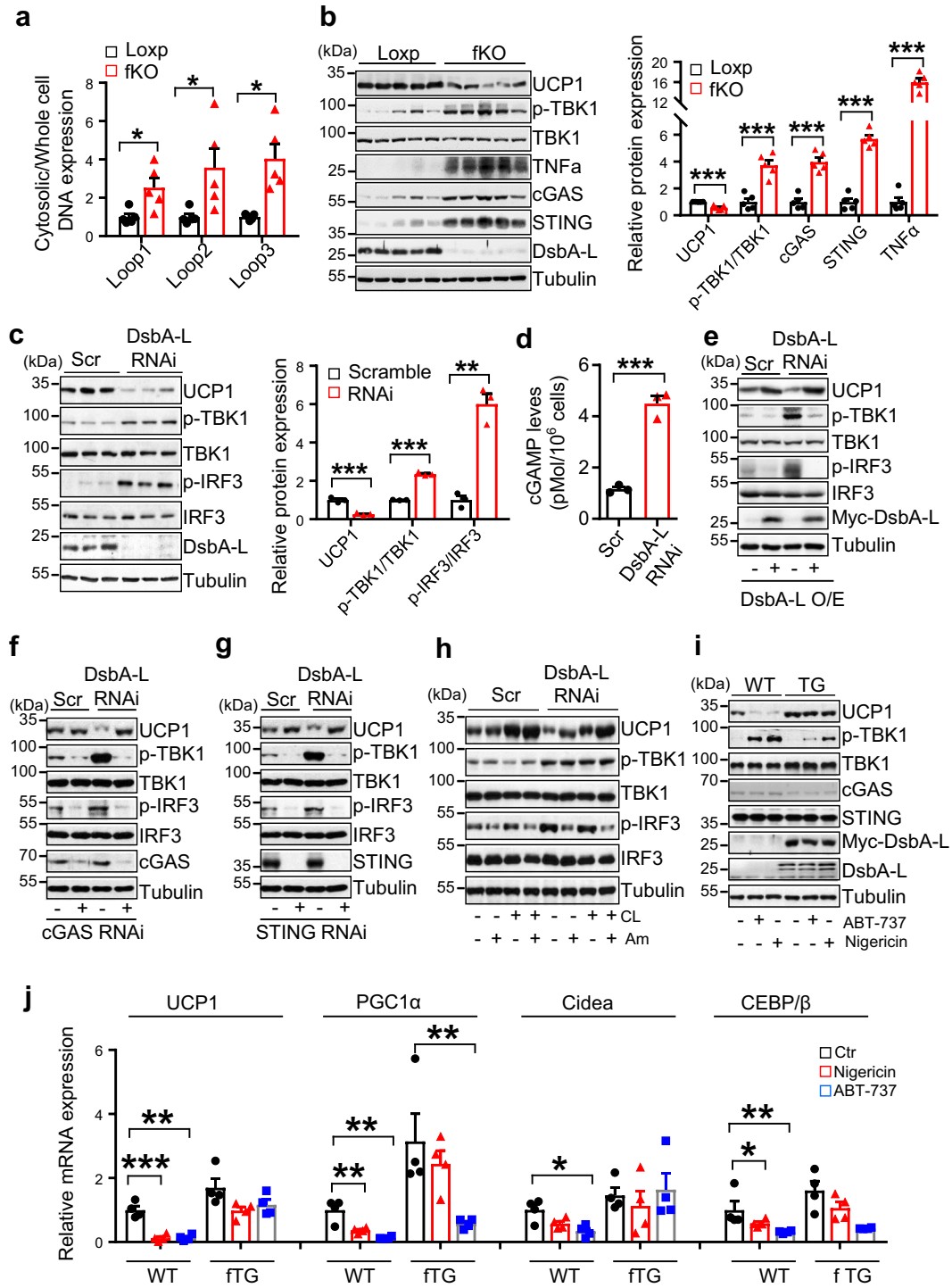

phosphorylation of TBK1 and IRF3 (Fig. 4e). To test whether the activation of the cGAS–STING pathway inhibits thermogenic gene expression, we suppressed cGAS or STING expression by RNAi in DsbA-L-deficient and control brown adipocytes. Suppressing cGAS (Fig. 4f) or STING (Fig. 4g) markedly reduced the phosphorylation of TBK1 and IRF3, concurrently with a great increase in UCP1 expression in DsbA-L-deficient cells. Moreover, treating DsbA-L-deficient brown adipocytes with amlexanox, a TBK1/IKKε specific inhibitor, significantly diminished IRF3 phosphorylation, accompanied by a significant increase in UCP1 expression with or without CL316243 treatment (Fig. 4h). On the other hand, the activation of the cGAS–STING pathway by

nigericin or ABT-737, two compounds known to stimulate mtDNA release[4,20], significantly suppressed thermogenic gene expression in primary brown adipocytes isolated from wild-type mice (Fig. 4i, j). The inhibitory effect of nigericin or ABT-737 on thermogenic gene expression was greatly attenuated in DsbA-L overexpressed cells (Fig. 4i, j). ABT-737 is a Bcl-2 inhibitor that has been found to promote caspase-dependent apoptosis and consequent cell death in MEFs and RAW264.7 cells[3,4,21]. Nigericin has also been shown to promote pyroptotic cell death[22,23]. To rule out the possibility that the reduced thermogenic gene expression in adipocytes is caused by increased cell death, we treated brown adipocytes, primary inguinal adipocytes, MEF, and

**Fig. 4 Activation of the cGAS–STING pathway mediates DsbA-L deficiency-induced inhibition of thermogenic gene expression. a** Cytosolic mtDNA content was quantitated via qPCR using mtDNA primers (Dloop1-3) in freshly purified brown adipocytes from DsbA-L$^{fKO}$ and loxp control mice ($n = 5$ for each group). **b** Immunoblot analysis of UCP1, STING, cGAS, TNFα, DsbA-L expression, and the phosphorylation of TBK1 at Ser$^{172}$ in BAT from DsbA-L$^{fKO}$ and loxp control mice ($n = 5$ for each group). The data were semiquantified by ImageJ program. **c** The protein levels of UCP1, DsbA-L, and the phosphorylation of TBK1 at Ser$^{172}$ and IRF3 at Ser$^{396}$ in DsbA-L-suppressed stable brown adipocytes and scramble control adipocytes ($n = 3$ for each group). The data were semiquantified by ImageJ program. **d** 2′3′-cGAMP levels in DsbA-L-suppressed brown adipocytes and scramble control cells were measured by HPLC–ESI–MS/MS analysis. **e** Representative immunoblot analysis of UCP1 expression, TBK1 and IRF3 phosphorylation in DsbA-L-suppressed brown adipocytes and scramble control cells stably overexpressing myc-tagged wild-type DsbA-L and its control plasmid. Representative immunoblot analysis of UCP1 expression, TBK1, and IRF3 phosphorylation in DsbA-L-suppressed brown adipocytes and scramble control cells transiently expressing **f** cGAS RNAi or **g** STING RNAi and their respective control plasmid. **h** Representative immunoblot analysis of UCP1 expression, TBK1, and IRF3 phosphorylation in DsbA-L-suppressed brown adipocytes and scramble control cells treated with or without 1 μM CL316243 or 50 μM amlexanox for 12 h. **i** Immunoblot analysis of UCP1, STING, cGAS expression, and TBK1 phosphorylation in primary brown adipocytes isolated from DsbA-L$^{fTG}$ and wild-type mice treated with or without 4 μM nigericin or 10 μM ABT-737 for 12 h. **j** mRNA levels of thermogenic genes in primary brown adipocytes isolated from DsbA-L$^{fTG}$ and wild-type control mice treated with or without 4 μM nigericin or 10 μM ABT-737 ($n = 4$ for each group) for 12 h. Data are presented as mean ± SEM of biologically independent samples, *$p < 0.05$, **$p < 0.01$, and ***$p < 0.001$ by unpaired two-tailed $t$-test (for comparison between two groups) or one-way ANOVA (for comparison of multiple groups).

RAW264.7 cells with ABT-737 or nigericin at different times. Noticeable cell death was observed in ABT-737- or nigericin-treated MEF and RAW264.7 cells, but the cell death rate was much lower in adipocytes compared with MEF and RAW264.7 cells (Supplementary Fig. 5a, b). Under these conditions, ABT-737 or nigericin greatly activated the cGAS–STING pathway as indicated by increased TBK1 or IRF3 phosphorylation, which correlated with a significant downregulation of UCP1 expression (Supplementary Fig. 5c, d). These results demonstrate that the ABT-737- or nigericin-induced downregulation of UCP1 in brown adipocytes is mainly due to activation of the cGAS–STING pathway rather than increased cell death.

There is some evidence suggests that ABT-737 is not sufficient to promote the cGAS–STING activation in the absence of the pan-caspase inhibitor Q-VD-OPh, because apoptosis-induced caspases cascade could silence the cGAS–STING-induced immune response[3,4,21]. Consistent with the findings of others[3,4,21], ABT-737 had no detectable effect on TBK1 or IRF3 phosphorylation in the absence of Q-VD-OPh in RAW264.7 or MEF cells (Supplementary Fig. 5c, d). However, we found that ABT-737 is sufficient to induce the cGAS–STING signaling in adipocytes, as demonstrated by increased TBK1 or IRF3 phosphorylation (Supplementary Fig. 5c, d). These results demonstrate a cell type-specific role of ABT-737 in promoting the cGAS–STING signaling pathway. Interestingly, whereas the stimulatory effect of ABT-737 on IRF3 phosphorylation in adipocytes could be further increased by the addition of Q-VD-OPh, UCP1 expression was not further reduced by Q-VD-OPh (Supplementary Fig. 5c, d), suggesting that cGAS–STING signaling may suppress thermogenesis via TBK1 but not IRF3 and the suppression is not due to the cell death.

**The cGAS–STING pathway inhibits adipocyte PKA signaling.** To elucidate the mechanism by which activation of the cGAS–STING pathway inhibits thermogenic gene expression, we examined the effect of cGAS–STING signaling on PKA activity, a key regulator of lipolysis, fatty acid oxidation, and thermogenesis in fat tissues[24]. Treating brown adipocytes with 2′3′-cGAMP, which stimulates STING-dependent TBK1 phosphorylation, inhibited PKA substrate phosphorylation, HSL phosphorylation, and UCP1 expression (Fig. 5a–c). In agreement with the finding that 2′3′-cGAMP acts downstream of cGAS but upstream of STING in the cGAS–STING pathway[25], the inhibitory effect of 2′3′-cGAMP on PKA signaling and UCP1 expression was greatly reduced in STING- (Fig. 5b), but not in cGAS- (Fig. 5c) suppressed brown adipocytes. Consistently, treating brown adipocytes with nigericin or ABT-737 inhibited PKA substrate phosphorylation and UCP1 expression in a time-dependent

manner (Fig. 5b, c and Supplementary Fig. 5e, f). Suppressing STING (Fig. 5b) or cGAS (Fig. 5c) by RNAi alleviated the inhibitory effect of nigericin or ABT-737 on PKA activity and UCP1 expression. To further determine the role of PKA signaling in the cGAS–STING pathway-induced inhibition of UCP1 expression, we treated cGAS- or STING-suppressed brown adipocytes or their respective control cells with the PKA inhibitor H89 in the presence or absence of 2′3′-cGAMP. The STING- or cGAS-deficiency-induced upregulation of UCP1 was inhibited by treating the cells with H89 (Fig. 5d, e), confirming a key role of PKA activation in the upregulation of UCP1 expression in the cGAS- or STING-deficiency cells. Inhibition of the cGAS–STING downstream kinases TBK1/IKKε by amlexanox in brown adipocytes also greatly rescued cGAMP-reduced PKA activity and downstream signaling including UCP1 expression, however, the effect was abolished when cells were treated with H89 (Fig. 5f). Taken together, these results revealed that the cGAS–STING pathway suppresses thermogenesis by inhibiting PKA activity and that the inhibition is mediated by TBK1/IKKε.

The mechanism by which the cGAS–STING pathway inhibits PKA activity remains unknown. We found that cAMP levels were much lower in DsbA-L-suppressed brown adipocytes compared with control cells, which were restored to a similar level to that of the control cells by treating the knockdown cells with 3′-isobutyl-1-methylxanthine (IBMX), a nonspecific PDE inhibitor, or zardaverine, a PDE3B and PDE4 selective inhibitor[26] (Fig. 5g). These findings suggest that activation of the cGAS–STING pathway may inhibit PKA activity by promoting PDE-mediated downregulation of cAMP. To test this possibility, we treated brown adipocytes with 2′3′-cGAMP. The treatment significantly reduced cAMP levels, which were greatly alleviated by incubating cells with IBMX or zardaverine (Fig. 5h). Moreover, knockout of STING or inhibition of PDE activity not only significantly increased basal- (Fig. 5i) and CL316243-induced (Fig. 5j) cAMP levels in primary adipocytes, but also greatly reduced the inhibitory effect of 2′3′-cGAMP on intracellular cAMP levels (Fig. 5i, j). Consistent with these results, 2′3′-cGAMP treatment greatly elevated PDE activity in primary adipocytes isolated from wild-type mice (Fig. 5k, l). However, the 2′3′-cGAMP-induced PDE activation was markedly suppressed in STING- but not cGAS-knockout cells (Fig. 5k, l). Collectively, these results demonstrate that the cGAS–cGAMP–STING pathway negatively regulates PKA signaling by activating PDE, thereby reduces thermogenic gene expression.

**Knockout of STING increased thermogenesis in mice.** To obtain direct evidence on the inhibitory role of the cGAS–STING pathway in regulating thermogenesis in vivo, we examined

cold-induced PKA signaling and UCP1 expression in adipose tissue of STING-deficient mice, also known as goldenticket mice (STING[gt]), which has been shown to be resistant to HFD-induced obesity[27,28]. Cold exposure-stimulated UCP1 expression, PKA activity, and HSL phosphorylation levels were significantly increased in both iWAT and BAT of STING[gt] mice compared with wild-type control mice (Fig. 6a, b). H&E staining revealed that cold exposure greatly reduced lipid droplets in BAT and markedly increased brown-like multilocular lipid droplet formation in iWAT of STING[gt] mice (Fig. 6c). Consistent with the

in vivo data, basal and CL316243-stimulated phosphorylation of PKA substrates including HSL and the expression of UCP1 were significantly increased in cultured primary inguinal adipocytes isolated from either STING[gt] mice or cGAS knockout mice (cGAS[−/−]) compared with their respective wild-type control mice (Fig. 6d, e). Treating primary inguinal adipocytes with 2′3′-cGAMP significantly inhibited CL316243-induced PKA signaling and UCP1 expression, however, this inhibitory effect was greatly abolished in primary adipocytes isolated from STING[gt] mice (Fig. 6d), further confirming a STING-dependent inhibition on

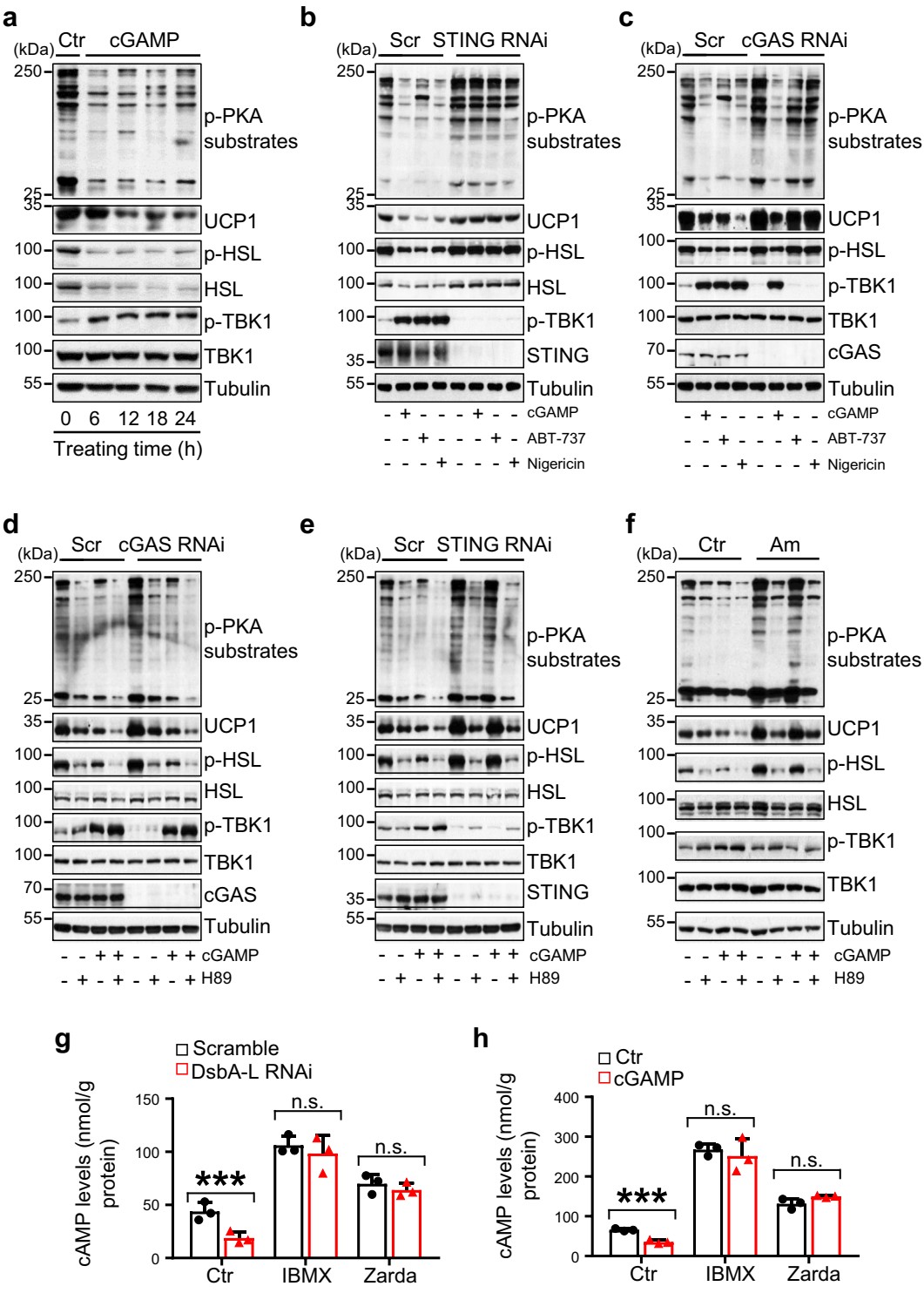

**Fig. 5** (continued)

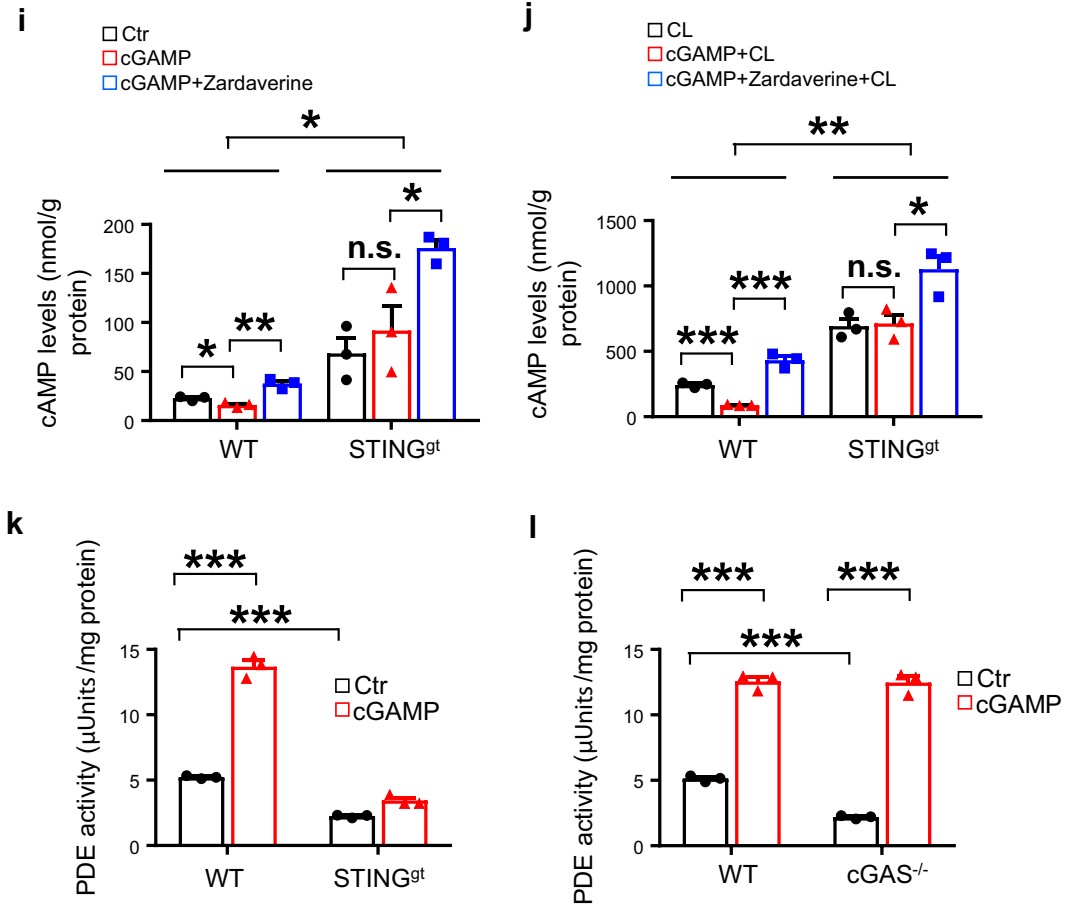

**Fig. 5 The cGAS–STING pathway inhibits cAMP-PKA signaling by activating PDE. a** Representative immunoblot analysis of UCP1 expression, and phosphorylation of PKA substrates, HSL, and TBK1 in brown adipocytes treated with 10 nM 2′3′-cGAMP at different time points as indicated. Representative immunoblot analysis of UCP1 expression, and the phosphorylation of PKA substrates, HSL and TBK1 in **b** STING RNAi or **c** cGAS RNAi brown adipocytes treated with 10 nM 2′3′-cGAMP, 10 μM ABT-737 or 4 μM nigericin for 12 h. Representative immunoblot analysis of UCP1 expression, and the phosphorylation of PKA substrates, HSL, and TBK1 in **d** cGAS-, **e** STING-, or **f** TBK1/IKKε-suppressed brown adipocytes treated with or without 10 nM 2′3′-cGAMP or 10 μM H89 for 12 h. **g** cAMP levels in scramble and DsbA-L-suppressed brown adipocytes treated with or without 250 μM IBMX or 10 μM zardaverine (Zarda) for 15 min (n = 3 for each group). **h** cAMP levels in brown adipocytes treated with or without 10 nM 2′3′-cGAMP for 6 h followed by treatment with or without 250 μM IBMX or 10 μM zardaverine (Zarda) for 15 min (n = 3 for each group). cAMP levels in primary adipocytes isolated from iWAT of wild-type and STING-deficient mice (STING^gt) (n = 3 for each group) treated with or without 10 nM 2′3′-cGAMP for 6 h followed by treatment of 10 μM zardaverine (Zarda) for 15 min in the **i** absence or **j** presence of CL316243. PDE activities were measured in primary adipocytes isolated from iWAT of **k** STING^gt or **l** cGAS^−/− mice (n = 3 for each group) treated with or without 10 nM 2′3′-cGAMP for 6 h. Data are presented as mean ± SEM of biologically independent samples, *p < 0.05, **p < 0.01, and ***p < 0.001 by unpaired two-tailed t-test (for comparison between two groups) or one-way ANOVA (for comparison of multiple groups).

PKA signaling. Concurrent with these findings, CL316243-induced lipolysis, as indicated by increased glycerol release, was markedly elevated in primary adipocytes isolated from STING^gt or cGAS^−/− mice compared with their respective control mice (Fig. 6f, g). In addition, the inhibitory effect of 2′3′-cGAMP on lipolysis was blocked in STING knockout cells at both basal and CL316243-stimulated conditions (Fig. 6f). On the other hand, overexpressing cGAS or STING in brown adipocytes greatly suppressed basal and CL316243-induced phosphorylation of PKA substrates, such as HSL, and the expression of UCP1 (Fig. 6h). Together, these results provide strong evidence that activation of the cGAS–cGAMP–STING pathway suppresses PKA signaling through a cell-autonomous mechanism. However, to our surprise, there was no significant difference in the expression of thermogenic genes between cGAS^−/− mice and wild-type control mice under either room temperature or cold exposure conditions (Supplementary Fig. 6a, b), suggesting that cGAS may have a more complicated role in regulating thermogenesis in vivo.

## Discussion

DsbA-L is a 25 kDa protein originally identified from the mitochondrial matrix[29]. We previously found that fat-specific knockout of DsbA-L promoted obesity in mice[5], but the underlying mechanism remains unclear. In the current study, we demonstrate that activation of the cGAS–STING pathway inhibits beige and brown fat thermogenesis, contributing to increased obesity in the DsbA-L^fKO mice. Conversely, fat-specific overexpression of DsbA-L or suppression of the cGAS–STING pathway promoted PKA signaling-dependent beige and brown fat thermogenesis, protecting mice from HFD-induced obesity. These findings demonstrate for the first time a negative role of the cGAS–STING pathway in regulating thermogenesis and energy expenditure, uncovering a potential therapeutic target to treat obesity and its related metabolic diseases.

Mitochondria, which are highly enriched in BAT and beige fat, play a critical role in nonshivering thermogenesis[30,31]. Mitochondrial dysfunction has been linked with impaired beige fat

development and thermogenesis[9,11,12]. However, the precise mechanism underlying mitochondrial dysfunction-induced inhibition of thermogenesis and the identity of a mitochondria-derived signaling pathway(s) remains largely unknown. An observation made in our study is that the mitochondrial stress-activated cGAS–STING pathway functions as a key sentinel signaling to suppress thermogenic program in adipose tissue (Figs. 4–6). It is well known that increased lipolysis and fatty acid oxidation in brown and beige adipocytes provide fuel for mitochondria to generate heat for the maintenance of body temperature in response to environmental changes such as cold exposure[32]. Dysfunction in mitochondria may generate a feedback signal to reduce fuel influx and thus shut down thermogenesis to prevent mitochondrial overloading. Taken together with the findings that mitochondrial stress activates the cGAS–STING pathway[16] and that activation of the cGAS–STING pathway inhibits thermogenesis, we speculate that mitochondrial stress-induced activation of the cGAS–STING pathway may provide an important feedback mechanism to prevent mitochondrial overloading, and thus alleviate mitochondrial dysfunction-induced cell damage. Nevertheless, chronic cold exposure has no significant effect on the cGAS–STING signaling pathway in adipose tissue (Supplementary Fig. 6a, b).

PKA is a key signaling node that controls β-3 adrenergic receptor-induced lipolysis, fatty acid oxidation, and thermogenesis[33]. Our study suggests that mtDNA stress-stimulated activation of the cGAS–STING signaling pathway may inhibit PKA signaling by promoting PDE activity. First, reduced cellular levels of cAMP and increased PDE activity were observed in DsbA-L-deficient or 2′3′-cGAMP-treated adipocytes, in which the cGAS–STING pathway was markedly stimulated (Fig. 5g–l). In addition, treating adipocytes with PDE inhibitors restored intracellular cAMP levels in DsbA-L-deficient or 2′3′-cGAMP-treated adipocytes (Fig. 5g–j). Furthermore, ablation of cGAS or STING greatly enhanced CL316243-stimulated cAMP-PKA signaling, lipolysis, and thermogenic gene expression, which is associated with decreased PDE activities in primary-cultured adipocytes (Figs. 5k–l, and 6d–g). These findings suggest that activation of the cGAS–STING pathway may inhibit PKA activity by activating PDE. The precise mechanism by which cGAS–STING pathway promotes PDE activity remains unclear, but we speculate that the increased of PDE activity in DsbA-L-suppressed or cGAMP-treated adipocytes may be caused by the activation of TBK1 or IKKε, two major downstream kinases of the cGAS–STING pathway[16,34,35]. In support of this hypothesis, we found that amlexanox-induced inhibition of TBK1/IKKε greatly rescued PKA activity and UCP1 expression in DsbA-L-suppressed or cGAMP-treated adipocytes (Figs. 4h and 5f). This result is consistent with the finding of pharmacologic inhibition of TBK1/IKKε by amlexanox elevated thermogenesis and energy expenditure in mice and IKKε or TBK1 could directly phosphorylate and activate the major adipocyte phosphodiesterase

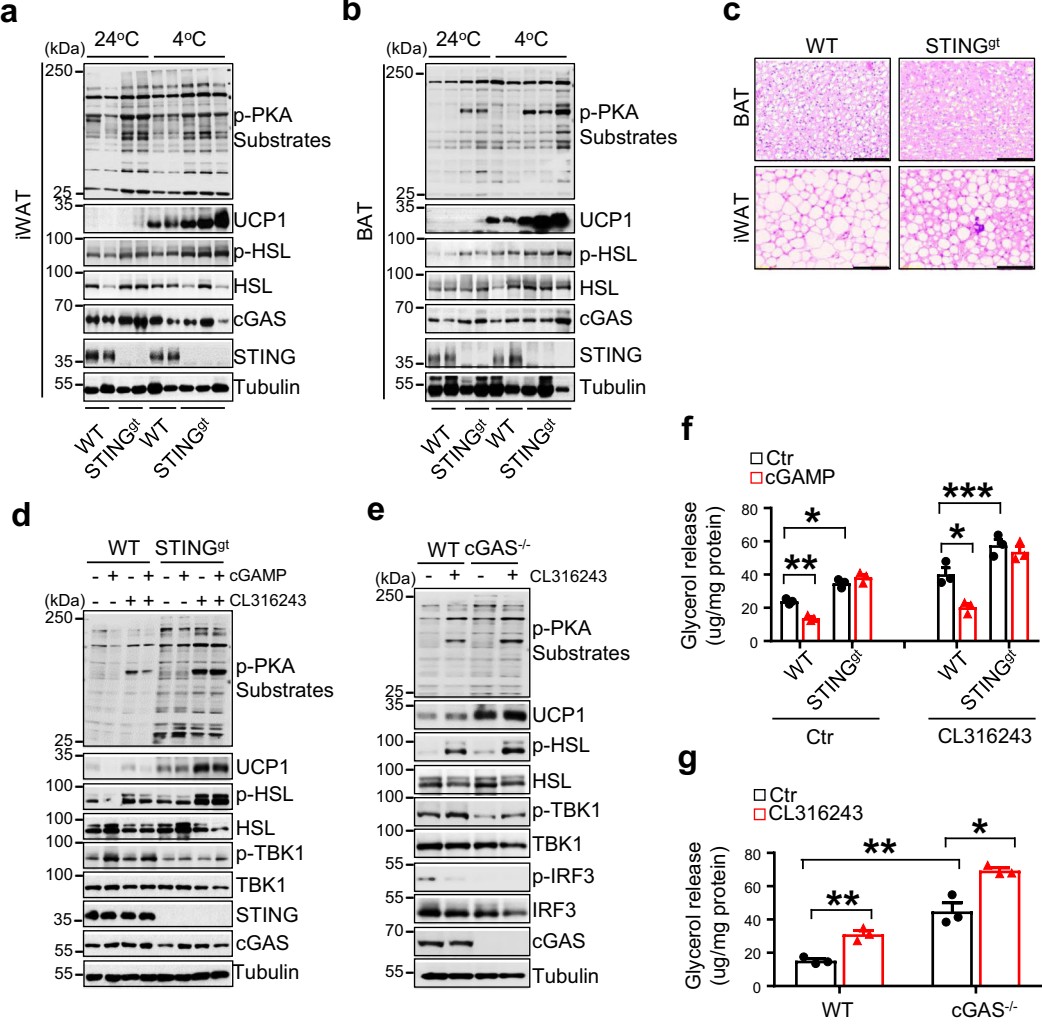

**Fig. 6** (continued)

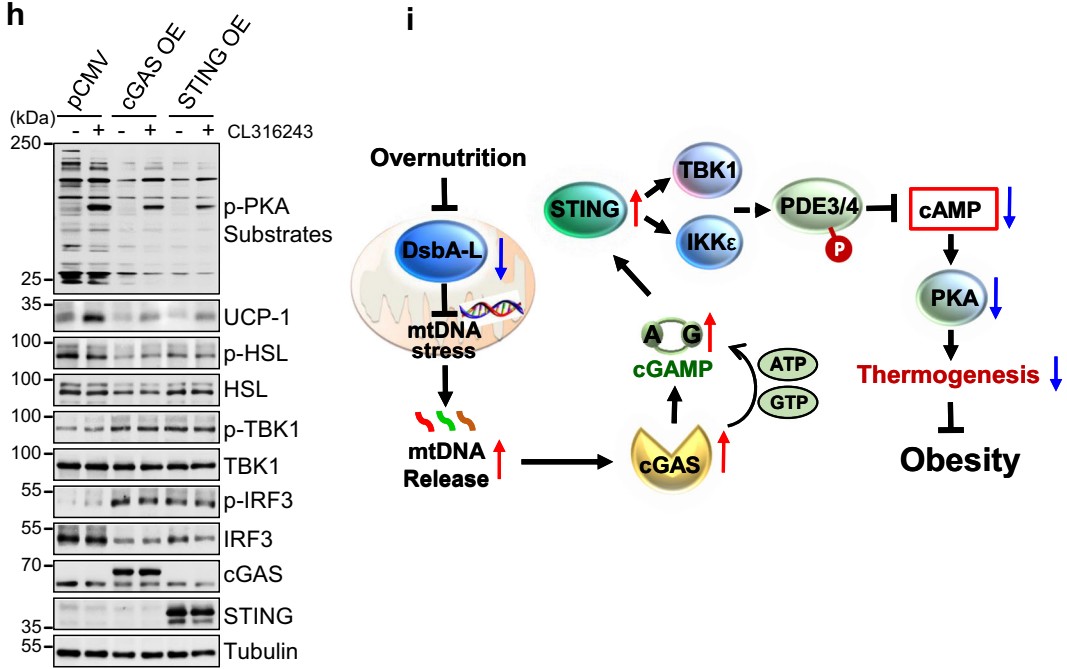

**Fig. 6 Knockout of cGAS or STING in mice increased PKA signaling and thermogenesis.** Immunoblot analysis of UCP1 expression and the phosphorylation of PKA substrates including HSL in **a** iWAT and **b** BAT of STING[gt] and their control littermates exposed to cold stress (4 °C) or housed at room temperature (24 °C). **c** Representative H&E stain of BAT and iWAT of wild-type and STING[gt] mice. Scale bar: 200 μM. Immunoblot analysis of UCP1 expression and the phosphorylation of PKA substrates, HSL, TBK1, and IRF3 in primary-cultured inguinal adipocytes treated with 1 μM CL316243 in the presence or absence of 10 nM 2′3′-cGAMP from **d** STING[gt] or **e** cGAS[−/−] mice and their wild-type control mice. Glycerol release from primary-cultured inguinal adipocytes treated with 1 μM CL316243 in the presence or absence of 10 nM 2′3′-cGAMP from **f** STING[gt] or **g** cGAS[−/−] mice and their wild-type control mice (n = 3 for each group). **h** Immunoblot analysis of UCP1 expression and the phosphorylation of PKA substrates, HSL, TBK1, and IRF3 in cGAS or STING overexpressed primary inguinal adipocytes treated with 1 μM CL316243 for 12 h. **i** A graphic model on the negative regulation of thermogenesis by the cGAS–STING pathway in adipose tissue. Data are presented as mean ± SEM of biologically independent samples, *$p < 0.05$, **$p < 0.01$, and ***$p < 0.001$ by unpaired two-tailed t-test (for comparison between two groups) or one-way ANOVA (for comparison of multiple groups).

PDE3B[36,37]. Together, these results reveal a potential biochemical mechanism by which activation of the cGAS–STING pathway inhibits thermogenic gene expression and energy expenditure.

While we found that knockout or suppression of cGAS or STING greatly increased CL316243-stimulated PKA substrates phosphorylation, lipolysis and UCP1 expression in cultured adipocytes (Figs. 5b, e and 6d–g), to our surprise, cold-induced thermogenesis was only observed in STING[gt] but not cGAS[−/−] mice (Supplementary Fig. 6a, b). These results suggest that cGAS may have a more complicated role in regulating energy homeostasis in vivo. As a key component of the innate immune system, cGAS functions as a cytosolic DNA sensor that stimulates STING-dependent inflammatory and interferon response[1,16]. However, this protein may also have a broad biological function in addition to activating the STING-dependent signaling events. cGAS has been found to localize in the plasma membrane[38] or the nucleus[39–41], where it may exert a cGAMP synthase activity-independent function. Therefore, while cGAS is negatively associated with PKA signaling pathway and UCP1 expression in adipocytes, it may play distinct roles in other adipose-resident cells, such as macrophages, that counter-regulate thermogenic gene expression in adipose tissue. Generation and characterization of cell-specific cGAS knockout mice should provide more information to answer these questions.

In summary, we show that activation of the cGAS–STING pathway, which inhibits PKA signaling via activation of PDEs, contributes to mitochondrial dysfunction-induced suppression of thermogenesis and energy expenditure in adipose tissue (Fig. 6i). Our findings shed insight into the mechanism by which

mitochondrial dysfunction feedback regulates thermogenic program. Increasing DsbA-L expression or suppressing cGAS–STING signaling in adipose tissue may be a potential therapeutic approach to ameliorate obesity-induced chronic inflammation and its associated metabolic diseases.

## Methods

**Materials**. CL316243, isoproterenol, resveratrol, rosiglitazone, amlexanox, zardaverine, H89, IBMX, and thiazolyl blue tetrazolium blue (MTT) were from Sigma. [1-$^{14}$C] palmitic acid was from Moravek Biochemicals. 2′3′-cGAMP and 2′3′-cGAMP control were from Invivogen. ABT-737, Q-VD-OPh and nigericin were from Cayman Chemical. Digitonin was from EMD Millipore. Antibodies to UCP1 (Cat# Ab23841), C/EBPβ (Cat# Ab32358), ChREBP (Cat #157153), Prdm16 (Cat# ab106410), and ERP57 (Cat# Ab10287) were obtained from Abcam; antibodies to cGAS (Cat# 31659), p-TBK1 (Cat# 5483), TBK1 (Cat# 3013), p-IRF3 (Cat# 4947), IRF3 (Cat# 4302), STING (Cat# 13647), PKA Substrates phosphorylation (Cat# 9621), p-HSL (Cat# 4139), HSL (Cat# 4107), PPARγ (Cat# 2443), ATGL (Cat# 2439), FASN (Cat# 3180), p-ACC (Cat# 3661), ACC (Cat# 3662), and SCD1 (Cat# 2794) were all from Cell Signaling; an antibody to PGC1α (Cat# ST1204) was from EMD Millipore; an antibody to TNFα (Cat# CG1601) was from Cell Applications, Inc; an antibody to Complex IV (Cat# A6403) was from Molecular Probes; Lamin A antibody (Cat# 3267-100) was from BioVision; antibodies to beta-tubulin and myc were made from monoclonal antibody-producing cell lines obtained from ATCC; A polyclonal anti-DsbA-L antibody is homemade from rabbit and also available in EMD Millipore (Cat# ABS1644).

**Animals**. All animal experiments were performed according to the procedures approved by the Institutional Animal Care and Use Committee at University of Texas Health San Antonio and performed in accordance with the Guidelines for the Care and Use of Laboratory Animals of the National Institutes of Health. Fat-specific DsbA-L knockout mice (DsbA-L[fKO]) and fat-specific DsbA-L transgenic mice (DsbA-L[fTG]) were generated as described previously[5]. For metabolic measurements, male DsbA-L[fKO] mice, DsbA-L[fTG], and their control littermates at 8 weeks of age were fed with either normal chow diet or 45% HFD (Research Diets

Inc, New Brunswick, NJ) for 16 weeks. For cold exposure experiments, 8 weeks old of male cGAS knockout mice (Jackson Laboratory; Stock Number: 026554), STING-deficient mice (STING[gt]) (Jackson Laboratory; Stock Number: 017537) were fed with normal diet and.

**Cells**. The brown preadipocyte cell line was a generous gift from Dr. Jiandie Lin (University of Michigan). DsbA-L stably suppressed or scramble control brown preadipocytes were generated using a small RNA interference approach according to a similar protocol as described previously[42]. Stable cell lines were generated by positive selection with G418, and DsbA-L knockdown was confirmed by western blot. Similar approach was used to generate stable cell lines in which a pSV2-hygromycin control plasmid and/or a plasmid encoding RNAi-resistant wild-type DsbA-L plasmid[43] were stably introduced in scramble control or DsbA-L stably-suppressed brown preadipocytes by positive election with hygromycin.

**Primary cell culture and differentiation**. Primary stromal vascular fractions (SVFs) and adipocytes from murine iWAT and interscapular brown fat depots were digested in isolation buffer containing 4% BSA and 1.5 mg/mL collagenase A (Roche). The cell suspension was filtered through a 100 μM filter and then centrifuged at $700 \times g$ for 3 min to separate floating adipocytes from the SVF pellet. Purified adipocytes were washed in PBS twice for further experiments. SVFs were cultured and differentiated to adipocytes as described previously[42].

**Energy expenditure measurement**. Energy expenditure of male DsbA-L[fKO] and Loxp control mice at 4 months of age was measured by metabolic cages according to the procedure as described previously[44]. Oxygen consumption ($VO_2$), carbon dioxide production ($VCO_2$), and the activity of each animal in live-in cages were measured for two complete light cycles and two complete dark cycles. Activity monitoring was performed simultaneously with metabolic measurements via the MAD-1 Motion/Activity Detector.

**Cold stress exposure and core body temperature measurement**. For cold-induced thermogenic gene expression analysis, individually housed male mice (3 months old) were kept at 4 °C for 6 h every day with free access to food and water continuously for 7 days. For cold tolerance studies, core body temperature of mice surgically implanted with the Mini-Mitter implantable bio-telemetric thermo-sensors was monitored using a telemetry system at various times of cold exposure[44]. Briefly, mice were individually housed with free access to food and water at room temperature (~24 °C) for 48 h, and then subjected to cold exposure (4 °C) for 6 h. The data were processed using the Vital View software.

**Lipolysis**. Lipolysis was performed according to the procedure as described[44]. In brief, differentiated adipocytes were incubated in 500 μL of KRB buffer containing 2% fatty-acid-free BSA and 0.1% glucose with or without 10 μM isoproterenol at 37 °C for 16 h. The KRB buffer were collected and used for fatty acid and free glycerol analysis using the NEFA C Kit (Wako) and Free Glycerol Reagent (Sigma), respectively. The levels of fatty acid and free glycerol were normalized to total protein levels in the cells.

**Fatty acid oxidation**. Fatty acid oxidation in adipocytes was determined by measuring $^{14}CO_2$ produced from oxidation of $^{14}C$-palmitate as described previously[44]. In brief, cells in 25 mm flasks were incubated at 37 °C with 1 mL medium containing 10 μM $^{14}C$-palmitate (53 mCi/mmol) and 20 μM L-carnitine for 30 min. After acidification with 200 μL 2.6 N $HClO_4$, $CO_2$ in the medium was trapped by hanging filter paper containing 30 μL 20% KOH in a sealed flask at 37 °C for 2 h. The trapped radioactivity was quantified by scintillation counting.

**Detection of mtDNA content in cytosolic extracts**. Cytosolic mtDNA content in cultured adipocytes or freshly purified mouse adipocytes was measured according to the procedure as described[2,5]. Total DNAs from one equal aliquot of cells were extracted and served as normalization controls for total mtDNA. mtDNAs were isolated from pure cytosolic fractions by using QIAQuick Nucleotide Removal Columns (QIAGEN). Quantitative PCR was performed on both whole-cell extracts and cytosolic fractions using nuclear DNA primers (Tert) and mtDNA primers (Dloop1-3, mtND4 and Cytb), and the CT values obtained for mtDNA abundance for whole-cell extracts served as normalization controls for the mtDNA values obtained from the cytosolic fractions. No nuclear Tert DNA was detected in the cytosolic fractions, indicating nuclear lysis did not occur.

**2'3'-cGAMP measurement (HPLC–ESI–MS/MS)**. 2'3'-cGAMP levels were measured on a Thermo Fisher Q Exactive LC-MS with a Thermo Fisher/Dionex Ultimate 3000 HPLC system according to the procedure as described in our previous studies[5]. The conditions used for 2'3'-cGAMP analyses were: Kinetex C18 column, (2.6 μm, 2.1 × 100 mm; Phenomenex); mobile phase A, 0.1% acetic acid with 10 mM ammonium acetate in water; mobile phase B, methanol; flow rate, 250 μL/min; gradient, held 1% B for 5 min and ran from 1% B to 30% B over 4 min. Full scan mass spectra were acquired in the orbitrap over an $m/z$ range of 100–1000 at 70,000

resolution ($m/z$ 300). Metabolite accurate mass (±5 ppm) was used for metabolite identification and agreed with the HPLC retention time of authentic standards. Quantification was made by integration of extracted ion chromatograms followed by comparison with the corresponding standard curves.

**Generation of cGAS and STING shRNA plasmid**. The shRNA sense and antisense sequences were chemically synthesized and ligated into the pRNATin-H1.2/Adeno vector (GeneScript). The sense strand sequences for m-cGAS and m-STING are 5'-GTGAGGACCAATCTAAGAC-3' and 5'-GAGCTTGACTCC AGCGGAA-3', respectively[5,45].

**Cellular cAMP measurement**. cAMP levels were measured using a Cyclic AMP ELISA Kit (Cayman). Briefly, differentiated adipocyte extract was isolated with 0.1 M HCl. Overall, 0.3–750 pmol/mL serial dilutions of the cAMP ELISA standards were used to establish standard curves. Fifty microliters of samples or standards were used for each reaction. ELISA buffer, AChE Tracer, and antiserum were added to the corresponding wells. The reaction was incubated for 18 h at 4 °C before rinsing five times with wash buffer. Overall, 5 μL of tracer was added to TA wells and 200 μL of Ellman's Reagent was added to all wells. The plate was developed for 120 min in the dark and read at 410 nm

**PDE activities**. PDE activities were measured using a Total PDE Activity Assay Kit (Fluorometric) (BioVision, Cat No: K927-100). Briefly, differentiated adipocytes were extracted with PDE assay buffer on ice. The Coumarin Standard reagent was diluted serially from 5 to 40 pmol/reaction to establish standard curves. The Reaction Mix, which contains PDE substrates, was added to the samples, positive control and background control. The Sample Background Control Mix, which only contains sample extract and assay buffer, was added to the samples background control. The 96-well microplate was read at Ex/Em 370/450 nm on kinetic mode by using a SynergyTM HT Multi-Detection Microplate Reader (BioTek Instruments, Inc).

**Real-time qPCR**. Total RNAs were isolated from homogenized tissue samples according to the manufacturer's suggested protocol. Quantitative PCR reactions were performed using SYBR green (Applied Biosystems) and quantitated using an Applied Biosystems 7900 HT sequence detection system or a CFX Touch-Real-Time PCR Detection System (Bio-Rad). To determine relative expression levels, duplicate runs of each sample were normalized to β-actin. The sequences for the primer pairs used in this study are listed in Supplementary Table 1.

**Statistics and reproducibility**. Statistical analysis was performed with GraphPad Prism 7 software. Statistical significance was determined by using one-way ANOVA or Student's unpaired $t$ test. All data are presented as mean ± SEM as indicated in the figure legends. Statistical significance was set at $*p < 0.05$, $**p < 0.01$, and $***p < 0.001$. For cell studies, data are representative of at least three independent experiments.

**Reporting summary**. Further information on research design is available in the Nature Research Reporting Summary linked to this article.

## Data availability
The source data underlying the graphs in figures are provided in Supplementary Data 1. The unprocessed gel blot images with size markers are provided in Supplementary Fig. 7. The authors declare that the data supporting the findings of this study are available within the article and from the corresponding author on reasonable request.

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

## Acknowledgements

We are grateful to Dr Jiandie Lin (Life Sciences Institute, University of Michigan) for his generous gift of brown adipocyte cell line. We also thank Dr Elizabeth Fernandez (Department of Pharmacology and Barshop Institute for Longevity and Aging Studies, UTHSA) for her kind help with the metabolic cage study. We also thank the efforts of the UTHSA institutional Mass Spectrometry Laboratory and support from NIH grant P30 CA54174. We also thank the support from Biobanking & Genome Analysis Core of UTHSA for real-time PCR experiment. This work was supported in part by NIH R01 grants DK114479, DK115761, and DK102965, Innovative Basic Science Awards of American Diabetes Association 1-19-IBS-147, and grants from the National Natural Science Foundation of China 81730022 and 81870601.

## Author contributions

J.B. and F.L. designed experiments. J.B., C.C., S.H., J.H., G.R.P., J.W., D.Y., C.Z., and M.L. performed experiments and analyzed the data. F.L., L.Q.D., and J.B. were involved in conceptualization and design, data analysis and interpretation, manuscript writing, financial support, and final approval of manuscript.

## Competing interests

The authors declare no competing interests.
