## [Peer Review File · Communications Biology]

Reviewers' comments:

Reviewer #1 (Remarks to the Author):

In their previous work, Bai et al reported that the fat-specific DsbA-L deficient mice suffer from obesity, metabolomics dysfunction, and chronic inflammation. One mechanism proposed there was that compromised mitochondrial integrity promoted mtDNA leakage to activate the cGAS-STING pathway. The current manuscript continues to describe their characterization of the fat-specific DsbA-L knockout mice and the underlying role of the cGAS-STING pathway. Here, they show that the DsbA-L-deficiency-related obesity may root in reduced heat generation, another possible consequence of the active cGAS-STING pathway. One possible way that the cGAS-STING pathway dampens thermogenesis was by activating phosphodiesterases to degrade cAMP thereby inhibit PKA activity. The phenotypic observation and characterization is of biological importance and interest. The mechanistic investigation would need to be strengthened with more evidence and clearer interpretation.

Line 27: Authors wrote in the abstract, "knockout of STING protects mice against high-fat diet-induced obesity..." However, no concrete data was found in the manuscript to directly support this important conclusion. Authors should provide evidence to show whether knockout of STING or cGAS may rescue the obesity phenotype, i.e. body weight, fat tissue, etc, in the fat-specific DsbA-L mice background, not just metabolomics or signaling defects in fat tissues. Otherwise, this sentence should be revised to fit the actual experiments described, and the significance of the study would be weakened. Besides obesity, authors should inspect thermogenesis in fat-specific DsbA-L and STING^{gt} double knockout mice. The proposed model would predict that STING or cGAS knockout rescue the thermogenesis defect.

Line 90: no data was found in the manuscript that compares food intake between DsbA-L KO mice and loxP littermates. Authors should show this data.

Line 121: "...DsbA-L has a cell autonomous effect on thermogenic gene expression." This interpretation is not accurate. The authors found that DsbA-L knockdown in stable adipose cells also led to reduced levels of thermogenesis proteins, mimicking tissues collected from mice. This may indicate that DsbA-L's function is tissue autonomous, not requiring other types of tissues or cells, not necessarily cell autonomous. Similar conclusions were made elsewhere and should be revised accordingly.

Line 183: ABT-737, a pan-BCL inhibitor, was used in this study to promote mtDNA release. ABT-737 can induce canonical apoptosis through mitochondria damage. Although it promotes mtDNA release as a co-lateral consequence, previous literatures show that ABT-737 by itself is insufficient to engage cGAS-STING signaling, due to the protection of caspase(s) (PMID: 25525874, 25525875). In fact, caspase inhibitor is required in conjunction to ABT-737 to induce robust cGAS-STING signaling. It is therefore surprising that the authors could only use ABT-737 to activate the cGAS-STING pathway, which is inconsistent with a large body of literature. Authors should also test the possibility that nigericin or ABT-737 regulate thermogenesis through their effect on cell death. The rate of cell death after treatment of ABT-737 or nigericin should be measured.

Figure 4-d and 4-f: measured cGAMP levels are incredibly high. Such high level of cGAMP is rare even for DNA transfected cells. It would help to show raw MS data and calibration curve to justify such high cGAMP levels, especially for the control cells in which cGAS is presumably not activated. 4-f: transient expression via plasmid transfection would directly activate cGAS, independent of mtDNA release. Stable overexpression of DsbA-L should be adopted here to reach meaningful conclusion.

Another minor point is about the word "retrograde" used in the manuscript to describe the regulation of mitochondria function onto thermogenesis. In definition, "retrograde" means "directed or moving back in space or time". Here "feedback" or "in a feedback manner" may be more suitable for describing such a logical connection in signaling pathway.

Reviewer #2 (Remarks to the Author):

In this study, the authors demonstrated a novel mitochondrial retrograde signaling pathway mediated by cGAS-STING in the control of adipose thermogenesis and energy homeostasis. Mitochondrial stress and dysfunctions are commonly associated with impaired adipocyte metabolism and insulin sensitivity; however, the mechanisms through which mitochondrial stress alters brown and beige fat thermogenesis remain unknown. The authors provided compelling evidence that supports a critical role of the cGAS-STING pathway in diminishing thermogenic response to mitochondrial stress. Using a combination of knockout and transgenic mouse models and cell culture studies, the authors demonstrated that cGAS-STING is an upstream regulator of PDE3/4 that suppresses cAMP/PKA response and thermogenic gene induction. A large body of high-quality data was included in this manuscript to support their conclusions. Overall, this is a very interesting study that provides new insights into adipocyte stress signaling and thermogenic regulation. Several points need to be addressed.

1. Defects in adipose tissue thermogenesis are often linked to increased lipid droplet sizes in brown fat and fewer multilocular adipocytes in iWAT. It would be helpful to include some adipose tissue histology data for the models used in this study.
2. Does fat specific knockout and transgenic expression of DsbA-L affect UCP1-independent thermogenic pathways. The authors can examine the expression of genes involved in calcium cycling and the creatine futile cycle.
3. Is the cGAS-STING pathway regulated by cold exposure?

Reviewer #3 (Remarks to the Author):

This manuscript investigates the effect of adipocyte DsbA-L, a chaperone-like mitochondrial localized protein, on thermogenesis in mouse adipose tissue. Using various animal models, three major observations were made: (1) adipocyte-specific DsbA-L knockout mice show decreased thermogenesis; (2) DsbA-L deficiency activates the cGAS-STING pathway in brown adipocytes; (3) activation of the GAS-STING pathway in brown adipocytes leads to reduced cellular cAMP levels, decreased PKA activity and suppressed thermogenesis. Their data uncovers a link between mitochondrial dysfunction-derived immune responses and energy expenditure in adipose tissue.

Overall there are a lot of nice data here and the authors should be commended for taking on this understudied area and doing a thorough job. There are both gain and loss of function data, and in vitro and in vivo models. The quality of the data look good. I have only a few minor concerns:

- 1) The authors should quantitate their western blots, which will increase the paper's readability.
- 2) The authors should include data addressing the effect of DsbA-L on brown or beige adipogenesis.
- 3) In Figure 6D, why are phospho-IRF3 levels so high in unstimulated WT control cells?

Responses:

Reviewer #1

Q1. *Line 27: Authors wrote in the abstract, “knockout of STING protects mice against high-fat diet-induced obesity...” However, no concrete data was found in the manuscript to directly support this important conclusion. Authors should provide evidence to show whether knockout of STING or cGAS may rescue the obesity phenotype, i.e. body weight, fat tissue, etc, in the fat-specific DsbA-L mice background, not just metabolomics or signaling defects in fat tissues. Otherwise, this sentence should be revised to fit the actual experiments described, and the significance of the study would be weakened. Besides obesity, authors should inspect thermogenesis in fat-specific DsbA-L and STING^{gt} double knockout mice. The proposed model would predict that STING or cGAS knockout rescue the thermogenesis defect.*

Response: We thank the reviewer for the constructive suggestion to generate fat-specific DsbA-L and STING^{gt} double knockout mice. Unfortunately, it is impractical for us to perform such study

given that it will take another year and half to generate and characterize the DsbA-L/STING^{gt} double knockout mice. In addition, we believe that the double knockout mouse model is not essential for the conclusion of our current study for several reasons. Firstly, we have already presented cell studies to show that suppressing cGAS, STING or TBK1/IKK ϵ in DsbA-L deficient adipocytes is sufficient to rescue DsbA-L deficiency-induced inhibition of UCP1 expression (Figs. 4f, 4g and 4h). In addition, knocking out STING has already been shown by others to protect mice against high-fat-diet-induced obesity (Luo X et al. (2018) *Gastroenterology*; 155:1971-198; Yu et al. (2019) *JCI*; 129:546-555. References 27 and 28 in our manuscript). Furthermore, the main focus of the current study is to determine the molecular mechanism by which activation of the cGAS-STING pathway accelerates overnutrition-induced obesity. Thus, we believe that lacking the STING^{gt} double knockout mouse data won't affect our conclusion that activation of cGAS-STING pathway in adipocytes inhibits thermogenesis by suppressing PDE3B-mediated downregulation of cAMP-PKA pathway.

Q2. *Line 90: no data was found in the manuscript that compares food intake between DsbA-L KO mice and loxP littermates. Authors should show this data.*

Response: We found that fat-specific knockout of DsbA-L had no significant effect on food intake in mice fed either a normal diet or a high-fat-diet and the results were presented at Supplementary Fig S2 A and B in our previous publication (Bai et al., *Proc Natl Acad Sci U S A.*, 2017, 114(46): 12196-12201). We have cited this work in the current study (Ref #5).

Q3. *Line 121: "...DsbA-L has a cell autonomous effect on thermogenic gene expression." This interpretation is not accurate. The authors found that DsbA-L knockdown in stable adipose cells also led to reduced levels of thermogenesis proteins, mimicking tissues collected from mice. This may indicate that DsbA-L's function is tissue autonomous, not requiring other types of tissues or cells, not necessarily cell autonomous. Similar conclusions were made elsewhere and should be revised accordingly.*

Response: We thank the reviewer for raising this point. However, given that a tissue or an organ usually contains multiple cell types and that our results showed that knockout or suppression of DsbA-L in cultured adipocytes alone is sufficient to inhibit thermogenic gene expression, we can only conclude that the effect of DsbA-L is cell rather than tissue autonomous.

Q4. *Line 183: ABT-737, a pan-BCL inhibitor, was used in this study to promote mtDNA release. ABT-737 can induce canonical apoptosis through mitochondria damage. Although it promotes mtDNA release as a co-lateral consequence, previous literatures show that ABT-737 by itself is insufficient to engage cGAS-STING signaling, due to the protection of caspase(s) (PMID: 25525874, 25525875). In fact, caspase inhibitor is required in conjunction to ABT-737 to induce robust cGAS-STING signaling. It is therefore surprising that the authors could only use ABT-737 to activate the cGAS-STING pathway, which is inconsistent with a large body of literature. Authors*

should also test the possibility that nigericin or ABT-737 regulate thermogenesis through their effect on cell death. The rate of cell death after treatment of ABT-737 or nigericin should be measured.

Response: To determine if the ABT-737- or nigericin-induced cell death contributes to reduced thermogenesis in adipocytes, we treated brown adipocytes, primary inguinal adipocytes, MEF, and RAW264.7 cells with ABT-737 or nigericin. We found that the cell death rate of adipocytes was much lower compared to that of MEF and RAW264.7 cells ((less than 10% in adipocytes compared to approximately 40% in MEFs and RAW264.7 cells at 4 and 12 hr treatment time points; Supplementary, Fig. 5a, b). Under these conditions, ABT-737 or nigericin greatly activated the cGAS-STING pathway as indicated by increased TBK1 or IRF3 phosphorylation, which correlated with a significant downregulation of UCP1 expression (Supplementary Fig. 5c, d). These results demonstrate that the ABT-737- or nigericin-induced down-regulation of UCP-1 in brown adipocytes is mainly due to activation of the cGAS-STING pathway rather than increased cell death.

The reviewer also pointed out that some studies showed that ABT-737 is insufficient to promote the cGAS-STING activation in the absence of the pan-caspase inhibitor Q-VD-OPh. Indeed, we found that ABT-737 had no detectable effect on TBK1 or IRF-3 phosphorylation in the absence of Q-VD-OPh in RAW264.7 or MEF cells (Supplementary, Fig. 5c, d). However, ABT-737 is sufficient to induce the cGAS-STING signaling in adipocytes, as demonstrated by increased TBK1 or IRF-3 phosphorylation (Supplementary, Fig. 5c, d). These results demonstrate a cell type-specific role of ABT-737 in promoting the cGAS-STING signaling pathway. Interestingly, whereas the stimulatory effect of ABT-737 on IRF-3 phosphorylation in adipocytes could be further increased by the addition of Q-VD-OPh, UCP1 expression was not further reduced by Q-VD-OPh (Supplementary, Fig. 5c, d), suggesting that cGAS-STING signaling may suppress thermogenesis via TBK1 but not IRF3 and the suppression is not due to the cell death.

Q5. *Figure 4-d and 4-f: measured cGAMP levels are incredibly high. Such high level of cGAMP is rare even for DNA transfected cells. It would help to show raw MS data and calibration curve to justify such high cGAMP levels, especially for the control cells in which cGAS is presumably not activated. 4-f: transient expression via plasmid transfection would directly activate cGAS, independent of mtDNA release. Stable overexpression of Dsba-L should be adopted here to reach meaningful conclusion.*

Response: We thank the reviewer for the question on the concentration of 2'3'-cGAMP in our study. The MS raw data were generated and analyzed by Dr. Xiaoli Gao, the Technical Director in our Institutional Mass Spectrometry Laboratory. Dr. Gao is an expert in mass spectrometry study but unfortunately, she has left our university two years ago. Nevertheless, we found that the 2'3'-cGAMP levels detected in our study ranging from $1\sim 5 \times 10^{-18}$ Mol/Cell (equivalent to $1\sim 5 \times 10$ pMol/ 10^6 Cell), which is lower or comparable to the levels of intracellular 2'3'-cGAMP reported by other labs (Carozza A, et al, doi: <https://doi.org/10.1101/539312>, 2019; Bose D, et al., Cell Chemical Biology, 2016, 1539-1549). For example, Bose et al. showed that the concentration of

2'3'-cGAMP per L929 cell is 6×10^{-17} Mol. Carozaa et al. showed 2'3'-cGAMP intracellular levels is around 2×10^7 molecules/cell, which is equivalent to 3.3×10^{-17} Mol/Cell. Therefore, we think the levels of 2'3'-cGAMP in our cell study is reasonable.

We thank the reviewer for the suggestion to use DsbA-L stable cells for the experiments. Based on this suggestion, we have generated adipocyte cells stably expressing DsbA-L (page 20, line 401-404). We found that stably overexpression of DsbA-L restored UCP1 expression in DsbA-L-deficient brown adipocytes, which is correlated with reduced phosphorylation of TBK1 and IRF3 (Fig. 4e).

Q6. *Another minor point is about the word “retrograde” used in the manuscript to describe the regulation of mitochondria function onto thermogenesis. In definition, “retrograde” means “directed or moving back in space or time”. Here “feedback” or “in a feedback manner” may be more suitable for describing such a logical connection in signaling pathway.*

Response: We thank the reviewer for this constructive comment and have made necessary changes in the revised manuscript (page 1, line 1 (Title); page 2, line 22, 31; page 3-4, line 59-60; page 4, 74; page 15, line 315; page 18, line 368).

Reviewer #2

Q1. *Defects in adipose tissue thermogenesis are often linked to increased lipid droplet sizes in brown fat and fewer multilocular adipocytes in iWAT. It would be helpful to include some adipose tissue histology data for the models used in this study.*

Response: As suggested by the reviewer, we have performed H&E staining experiment, which revealed that adipose tissue-specific knockout of DsbA-L greatly suppressed cold-induced formation of small and multilocular lipid droplets beige adipocytes in mouse iWAT and increased lipid accumulation in BAT (Fig. 2e). We also observed an increased brown-like multilocular lipid droplet formation in iWAT and greatly reduced lipid droplets in BAT of STING^{gt} mice compared to their wild-type mice (Fig. 6c). These new data further support our conclusion that DsbA-L deficiency-induced activation of cGAS-STING pathway negative regulates brown fat thermogenesis and beige fat formation.

Q2. *Does fat specific knockout and transgenic expression of DsbA-L affect UCP1-independent thermogenic pathways. The authors can examine the expression of genes involved in calcium cycling and the creatine futile cycle.*

Response: We thank the reviewer for this constructive suggestion. Based on this suggestion, we have performed new experiments. We found that cold exposure had a comparable stimulatory effect on the mRNA expression of calcium cycle-related gene *serca2b* and creatine metabolism-related genes including *ckmt1*, *ckmt2*, *gatm* in iWAT of both the loxp control mice and DsbA-L^{fKO} mice

(Supplementary, Fig. 2a, b, c, d), indicating DsbA-L deficiency in adipose tissue had no significant effect on these UCP1-independent mechanisms underlying cold-induced beige fat thermogenesis.

Q3. *Is the cGAS-STING pathway regulated by cold exposure?*

Response: We found that cold exposure has no significant effect on the cGAS-STING signaling pathway in adipose tissues, at least under our chronic cold exposure conditions (Supplementary, Fig. 6a, b).

Reviewer #3

Q1. *The authors should quantitate their western blots, which will increase the paper's readability.*

Response: We thank the reviewer for this constructive suggestion. As suggested by the reviewer, we did quantification for some of the western blots to further demonstrate the difference in either phosphorylation or protein expression differences (Fig. 2f, g, h; Fig. 4b, c; Supplementary, Fig. 5c, d). However, we did not quantify all of the Western blot data because either the differences in those Western blots were so obvious or that we did not attempt to compare the difference between treatment groups in those experiments.

Q2. *The authors should include data addressing the effect of DsbA-L on brown or beige adipogenesis.*

Response: We thank the reviewer for this constructive suggestion. We measured adipogenesis marker FABP4 mRNA levels in BAT and iWAT and found that there is no significant difference between loxp control mice and DsbA-L fat KO mice (Supplementary, Fig. 3h).

Q3. *In Figure 6D, why are phospho-IRF3 levels so high in unstimulated WT control cells?*

Response: To show the phospho-IRF3 levels in all cells, we previously used the WB with the highest exposure time. To avoid misleading, we replaced this WB with a new one which has a lower exposure time in the revised manuscript (Fig. 6e).

REVIEWERS' COMMENTS:

Reviewer #1 (Remarks to the Author):

The authors have addressed most of my critics and concerns. The double knockout experiment would have added a lot more strength to the revised manuscript. However, I understand that the genetic experiment is time-consuming and perhaps not absolutely required.

Reviewer #2 (Remarks to the Author):

The authors have sufficiently addressed my comments.

Reviewer #3 (Remarks to the Author):

I have no further issues.